# Telementoring in Minimally Invasive Esophageal Atresia Repair: Results of a Case-Control Study and Lessons Learned from the TIC-PEA Study (Telemedical Interdisciplinary Care for Patients with Esophageal Atresia)

**DOI:** 10.3390/children9030387

**Published:** 2022-03-10

**Authors:** Tatjana Tamara König, Maria-Christina Stefanescu, Emilio Gianicolo, Anne-Sophie Holler, Oliver J. Muensterer

**Affiliations:** 1Department of Pediatric Surgery, Universitätsmedizin, Johannes-Gutenberg University, 55131 Mainz, Germany; christina.stefanescu@unimedizin-mainz.de; 2Institute of Medical Biostatisics, Epidemiology and Informatics (IMBEI), Universitätsmedizin, Johannes-Gutenberg University, 55131 Mainz, Germany; emilio.gianicolo@uni-mainz.de; 3Institute of Clinical Physiology, National Research Council, Campus Universitario Ecotekne, 73100 Lecce, Italy; 4Department of Pediatric Surgery, Dr. von Haunersches Kinderspital, Ludwig Maximilian-University Munich, 80337 Munich, Germany; annesophie.holler@med.uni-muenchen.de (A.-S.H.); oliver.muensterer@med.uni-muenchen.de (O.J.M.)

**Keywords:** esophageal atresia, telemedicine, telementoring, minimally invasive surgery, thoracoscopy

## Abstract

Minimally invasive esophageal atresia (EA) repair is deemed one of the most demanding procedures in pediatric surgery. Open repair is considered the gold standard and learning opportunities for minimally invasive repairs remain scarce. “Telemedical Interdisciplinary Care for Patients with Esophageal Atresia (TIC-PEA)” offers free access to an interdisciplinary network of experts for telemedical consultation (telementoring). The aim of this study was to determine the frequency of minimally invasive surgery (MIS) in TIC-PEA patients compared to the general population. TIC-PEA patients were matched and compared to controls regarding the use of MIS, patient characteristics, and complications. Patients (*n* = 31) were included at a mean age of 62.8 days (95%-CI: 41.4–84.3, 77% after the primary esophageal repair). The odds-ratio to have MIS was 4.03 (95%-confidence interval: 0.79–20.55) for esophageal anastomosis and 4.60 (95%-confidence interval: 0.87–24.22) for tracheoesophageal fistula-repair in the TIC-PEA group. Telementoring offered the chance to select the ideal candidate for MIS, plan the procedure, and review intraoperative images and videos with the expert. Telementoring as offered is ideal to promote MIS for EA and helps to address the individual learning curve. In order to maximize benefits, patients need to be included prior to the first esophageal procedure.

## 1. Introduction

Esophageal atresia repair is one of the most challenging minimally invasive procedures in pediatric surgery. After the first successful thoracoscopic repair in 1999 [1], minimally invasive surgery (MIS) has evolved in terms of operative equipment and technique. In a survey from 2014, only 6% of senior surgeons preferred the minimally invasive approach [2]. Nevertheless, in 2021, thoracoscopic repair was considered a viable option in the European Reference Network on Rare Inherited and Congenital Anomalies (ERNICA) consensus conference, if suitable expertise is available [3]. Some centers report 15.6% [4] up to 34.7% [5] minimally invasive repairs, with liberal conversion-rates of 33.3% [5] to 53.6% [4]. However, esophageal atresia (EA) and tracheo-esophageal fistulae (TEF) are rare malformations. Before reaching the plateau of the learning curve, less seasoned surgeons produce a 15% higher leakage rate and a 19–77% higher stenosis rate, compared to experienced surgeons [6]. Because amenable cases are rare, it is hard for a single surgeon to reach a plateau in the learning curve in a foreseeable time. In order to prepare surgeons for this procedure, mentoring programs have been implemented [7]. A structured program may include theoretical and experimental training, simulation, training in centers of reference, and personal operative experience [7]. Personal mentoring is a crucial part of this program [7]. In addition to personal consultation, mentoring may be performed via telemedical professional exchange. “Telemedical Interdisciplinary Care for Patients with Esophageal Atresia (TIC-PEA)” is a platform that allows for digital pre-, peri-, intra-, and post-operative telemedical consultation with experts in esophageal atresia and minimally invasive surgery. In this setting, minimally invasive surgery has a major advantage, in that intraoperative images and video clips, as well as live imaging data, can be evaluated by the mentor. This nationwide network, in which practitioners can include and register their patients, gives them access to expert opinions and support.

The aim of study is to compare the proportion of minimally invasive EA repair in the study cohort to the general population, and to explore the technical requirements, feasibility, and possibilities of telemedical support for minimally invasive surgery.

## 2. Materials and Methods

### 2.1. Telemedical Interdisciplinary Care for Patients with Esophageal Atresia

TIC-PEA was implemented in August 2020 as a free telemedical resource for multidisciplinary expert opinion in esophageal atresia care. Experts are chairpersons of German academic centers experienced in the treatment of esophageal atresia and/or members of the advisory board of the KEKS e.V. (Kinder und Erwachsene mit kranker Speiseröhre, the German national patient support group). After receiving informed consent from both caregivers and practitioners, infants born with esophageal atresia may be included throughout the first year of life. Ideally, infants are included prior to esophageal atresia repair. Telemedical consultations are continued until the first birthday. Standardized, regular consultations are planned upon presentation, before the operation, after the operation, and at regular follow-up intervals. Individual cases are consulted by the same expert, and more or multidisciplinary experts are included if specific questions arose. Intraoperative consultations are also possible upon request. Individual criteria to determine whether MIS is advisable include the level of the surgeons’ experience, local infrastructure, type of procedure (esophageal elongation or isolated closure of the fistula, or anastomosis) and patient characteristics (weight, type of esophageal atresia, and co-morbidity).

Universally-available hardware in the form of smartphones, tablets, laptops, or desktop computers are utilized for the interaction. Software guarantees data privacy and security.

For individual telemedical consultations, technical data and shared use of patient images and videos during the conferences is recorded and statistically analyzed.

Patient data on comorbidity, operative procedures, and complications are recorded according to standardized protocols. Here, we describe our experience with the platform and project.

### 2.2. Patients

Infants included in the TIC-PEA study between 1 August 2020 and 31 December 2021 were included in the study (TIC-PEA group). The TIC-PEA group was compared to historic controls of the voluntary EA patient registry courtesy of KEKS e.V. Data from 2010 through 2021 (control group) were included. Participants of the TIC-PEA study were excluded from the control group.

For the match-control-analysis, combined EA/TEF (Gross Type B/C/D) and patients with isolated EA (Gross Type A) were analyzed separately due to differences in incidence and surgical approach.

Cases included in the TIC-PEA cohort were stratified according to Gross Type. Two controls for each case (Gross Type B/C/D) were extracted from the control group. Infants born after 1 January 2017 and with information provided on birth weight were considered for the matched analysis. Controls were individually matched to cases for birth weight ±200 g. For Gross Type A, no formal match-control analysis was possible. Therefore, a descriptive analysis was performed.

### 2.3. Statistical Analysis

For continuous variables, means and 95%-confidence intervals (95%-CI) were calculated. A difference with non-overlapping 95%-CI was considered statistically significant. Conditional logistic regression models were used to calculate the odds ratio (OR) for primary minimally invasive anastomosis and minimally invasive TEF-closure, assuming patients of the control group as the reference. Odds ratios were adjusted for birth weight. The analysis was conducted with SAS software version 9.4 (SAS Institute, Inc., Cary, NC, USA).

## 3. Results

### 3.1. Telemedical Consultations

Overall, 81 telemedical consultations on 31 patients from 17 pediatric surgery centers (10 academic and 7 non-academic centers) were included. In three centers, the hospital firewall prohibited the use of all hospital hardware. Other than that, problems could be resolved in due course by adapting the software settings. Consultations were conducted by one out of six experts from high-volume centers, and lasted 14.7 min (95%-CI: 12.0–17.4 min). All patients were discussed by oral account, and in 15 patients, radiological findings were shared on the screen to be discussed. Furthermore, endoscopic findings (images *n* = 3, videos *n* = 3) were reviewed. In seven patients, minimally invasive procedures were discussed preoperatively, and intraoperative images (*n* = 3) or videos (*n* = 2) were reviewed in the postoperative debriefing.

### 3.2. TIC-PEA Patients

A total of 31 patients were included in the TIC-PEA group (Gross Type C *n* = 24, Gross Type A *n* = 6, and Gross Type D *n* = 1). No patient with Gross Type E was included. Almost one third (*n* = 9) were classified as long-gap EA by the primary surgeon. Comorbidities were present in 15 patients (VACTERL association *n* = 8). The mean patient age at the first telemedical consultation was 62.8 days (95%-CI: 41.4–84.3). Most patients (77%) were included after the first esophageal procedure had been performed. The mean follow-up after esophageal repair was 207.7 days (59–412 days). Of the seven patients included before esophageal repair, six had a long-gap EA and one was included by a surgeon who had included other patients before. Six of these cases had minimally invasive surgery (thoracoscopic elongations *n* = 5, thoracoscopic anastomosis *n* = 1), so the eligibility criteria and technical aspects for MIS were discussed preoperatively. One patient in this group had a complication associated with minimally invasive repair, and one complication occurred after open anastomosis.

Of the 24 patients included after the first esophageal procedure, most (*n* = 21) had primary esophageal anastomosis and TEF-repair (open repair *n* = 16, thoracoscopic repair *n* = 5). Two children with EA without TEF had esophageal anastomosis only (open repair *n* = 1, thoracoscopic repair *n* = 1). Complications associated with primary esophageal repair occurred in 13 patients (10 after open and 3 after minimally invasive repair) and included anastomotic stenosis (*n* = 10), anastomotic leakage (*n* = 2), one re-occurring TEF, and four other early complications. One patient died. In these cases, mainly complication management was addressed during the telemedical consultations. To date, no intraoperative live-mentoring for MIS has been performed.

### 3.3. KEKS e.V. (Control) Group

In the control group, data regarding the approach of primary repair (open vs. minimally invasive) were available for 258 patients with Gross Type B/C/D (primary esophageal anastomosis and TEF-repair *n* = 242). The mean birth weight was 2749 g (95%-CI: 2531–2968 g, *n* = 200) for patients who had open and 2762 g (95%-CI: 2553–2971 g, *n* = 33) for patients who had minimally invasive repair (data on birth weight missing *n* = 9). There was a fluctuating number of minimally invasive repairs over the study period of 11 years. No increase of minimally invasive repair over time could be shown (Figure 1). The mean overall percentage of minimally invasive repairs was 14.0% (95%-CI: 10.8–17.1). All isolated TEF-repairs (*n* = 16) were performed via thoracotomy at a mean weight of 2108 g (95%-CI: 1493–2722 g, *n* = 15, missing data *n* = 1). For controls with Gross Type A, no data on the approach to esophageal repair (open vs. minimally invasive) were provided.

### 3.4. Matched Analysis of Patients with EA/TEF (Gross Type B/C/D)

In the match-control analysis, 24 TIC-PEA patients with EA/TEF (Gross Type B/C/D) were matched with 48 randomly selected controls. TIC-PEA patients had a higher probability of having minimally invasive esophageal repair (anastomosis OR 4.03, 95%-CI: 0.79–20.55, TEF-repair OR 4.60, 95%-CI: 0.87–24.22) compared to the controls (Figure 2). Patients and controls had a comparable mean birth weight and gestational age. TIC-PEA patients had higher rates of associated malformation (Table 1). Primary repair was performed on the second day of life in both groups. In the TIC-PEA group, more patients had comorbidity, an isolated TEF-repair and gastrostomy, and more overall complications (Table 1).

### 3.5. Analysis of Patients with Isolated EA (Gross Type A)

Six patients with isolated EA (Gross Type A) were included both in the TIC-PEA group and in the control group (born 2017–2021). Even though TIC-PEA patients had a higher mean birth weight and gestational age, the difference was not statistically relevant (Table 2). In both groups, surgical management was heterogenous (Table 2). In the TIC-PEA group, four patients had minimally invasive and one had open elongation procedures. One patient had a gastric transposition as definitive repair after elongation. For the controls, no data regarding open or minimally invasive procedures were provided. Mean time to anastomosis was 110.0 days (95%-CI: 75.6–144.4) in the TIC-PEA group and 147.3 days (95%-CI: 87.9–206.6, *n* = 4, missing data *n* = 2) in the control group.

## 4. Discussion

Our data show that telementoring through TIC-PEA was associated with a higher proportion of minimally invasive EA repair, despite a higher rate of comorbidities and therefore complexity of the patients. In TIC-PEA patients, a minimally invasive approach was used four times as often compared to the controls. In the control group, only 14.0% (95%-CI: 10.8–17.1) of esophageal repairs were performed as minimally invasive in EA/TEF (Gross Type B/C/D) over the past 11 years, independent of birth weight. No change in this rate could be shown for the more recent years (Figure 2). As the vast majority of patients already had esophageal surgery at the time of inclusion, TIC-PEA conferences had no impact on the decision regarding open or minimally invasive repair in most patients. However, only few centers have more than one or two surgeon experienced in minimally invasive EA repair. In order to get in touch with other experienced surgeons, TIC-PEA offers ideal conditions for digital networking.

Combined with minimally invasive procedures, there is a greater profit from telementoring compared to open surgery. The digital video or image documentation of operations in MIS offers perfect conditions for telementoring [8]. Many pre-, intra-, and postoperative factors contributing to successful minimally invasive esophageal atresia repair can be addressed synchronously or asynchronously via telemedicine. Criteria concerning the patient, the surgeon, and the local setting and back-up resources should be considered. Preoperative patient selection is crucial for postoperative outcome. Generally accepted selection criteria include EA Gross Type C, birth weight >2000 g, absence of major associated malformations, and cardio-respiratory stability [7]. However, a recent study comparing the outcome after minimally invasive esophageal atresia repair in a high-volume center in infants with a body weight below and over 2000 g, found there was no difference in early postoperative complications [9]. Therefore, standard criteria might not be applied in all cases and individual counselling is warranted. Specific anticipated intraoperative difficulties can be discussed ahead of the operation. Reasons for conversion to open surgery include adverse events, anastomosis under tension, or lack of operative progress for more than 15 min [10]. Synchronous telementoring with real time video supervision and two-way audio communication for minimally invasive surgery was first published in 1996 [11]. There was no increase in operating time or the number of adverse events compared to mentoring by a senior surgeon in the operating room [11]. In pediatric surgery, remote site mentoring has been reported for many minimally invasive procedures, but not esophageal atresia [8].

In TIC-PEA conferences, intraoperative images and even videos from minimally invasive repair were reviewed without technical difficulties. In this way, the expert was able to objectively assess the procedure, instead of having to rely on the surgeon’s subjective oral account. Even though the intervention aims to include infants before the esophageal repair, and discuss patient characteristics and operative approach in order to help reduce complications, the majority of infants were included after esophageal repair or even only after complications occurred. In these cases, strategies for complication management were discussed. Compared to asynchronous consultations, live-coaching requires advanced technical and software equipment and careful planning. Telemedically supervised minimally invasive esophageal repair was planned for TIC-PEA patients, but has not been requested by the participating surgeons to date. This strategy might be of use to surgeons, who are technically experienced in minimally invasive techniques themselves, but like to have a second expert opinion on the individual case in order to address the learning curve.

In our series, a disproportionately high number of complicated cases with comorbidity (51.6%) or even VACTERL association (25.8%), EA without fistula (Gross Type A, 19.4%), and long-gap EA (29.0%) were included. These patients need an individually tailored surgical and multidisciplinary concept to achieve esophageal continuity. As, in the majority of those cases, no primary repair during the first days of life is possible, they are ideal candidates for telemedical mentoring to plan, execute, and review the surgical process of staged repair, and include anesthesiologists and neonatologists to develop an individual multidisciplinary concept. In the small group of patients with isolated EA (Gross Type A), esophageal continuity was achieved earlier in TIC-PEA-patients compared to controls. For patients with long-gap EA, the vast majority of elongation procedures were performed with a minimally invasive (*n* = 4) compared to an open (*n* = 1) or endoluminal (*n* = 1) approach.

Telemedical mentoring in surgery has been used successfully for more than 30 years, especially for minimally invasive procedures [11]. In contrast to most telemedical programs, TIC-PEA does not focus on the patient–specialist interaction (teleconsultation) but on the specialist–specialist interaction (telementoring [8]). Technical requirements to participate in the TIC-PEA study include a stable internet connection, and a device running a compatible web browser, camera, and microphone. Even though most private devices, such as a smartphones or tablet computers, meet these requirements, the electronic data processing systems of participating hospitals do not routinely allow universal access. Strict data policy in Germany prohibits internet access for devices with a connection to the medical information system containing patient information. The local operating site firewall may block the webservice or any microphone or camera for hospital devices. A lack of standardized hardware and software prohibited video conferences from official hospital devices in three participating centers due to data privacy issues or firewall settings. In order to guarantee a successful conference, participants should be advised to test hardware and software before the meeting.

Even though the shut down during COVID-19 pandemic sped up the process of digitalization in many areas [12], this technical potential could not be utilized in the TIC-PEA study to the fullest extent. In order for patients and caregivers to get the maximum benefit from the intervention, infants need to be included in the study before esophageal repair. In this way, a pre-/intra- and postoperative multidisciplinary assessment and planning might help to improve surgical outcome.

In Germany, the treatment of esophageal atresia is not centralized. Over the period of 2015–2018, 65.8% of patients were treated in small centers (<4 cases/year) and one third (34.2%) in medium centers (4–9 cases/year)—no patient was treated in a high-volume center with ten or more yearly cases [13]. Nevertheless, the concept of the TIC-PEA study has been controversial among pediatric surgeons in Germany from the beginning. Opponents are concerned that surgeons with insufficient expertise are encouraged to perform procedures surpassing their own surgical skills, instead of transferring patients to more experienced centers. We would like to emphasize that telementoring as practiced cannot replace formal surgical training. It aims to support care for patients with rare diseases by providing a national digital network to receive feedback on treatment plans, discuss possible pitfalls, or even transfer patients, if deemed necessary.

## 5. Conclusions

Minimally invasive esophageal atresia repair in Germany is still the exception. In addition to a low overall caseload, optimal patients and setting for minimally invasive repair is rarer still and must be reviewed carefully. Even surgeons experienced in minimally invasive surgery need a safety network for thoracoscopic esophageal atresia repair. The telementoring, provided by the TIC-PEA program, including ideal patient selection, review of the surgical strategy, and postoperative debriefing, provides additional safety for patients with esophageal atresia. In order to use it to its full potential, telementoring must commence before the first surgical steps are taken. 

## Figures and Tables

**Figure 1 children-09-00387-f001:**
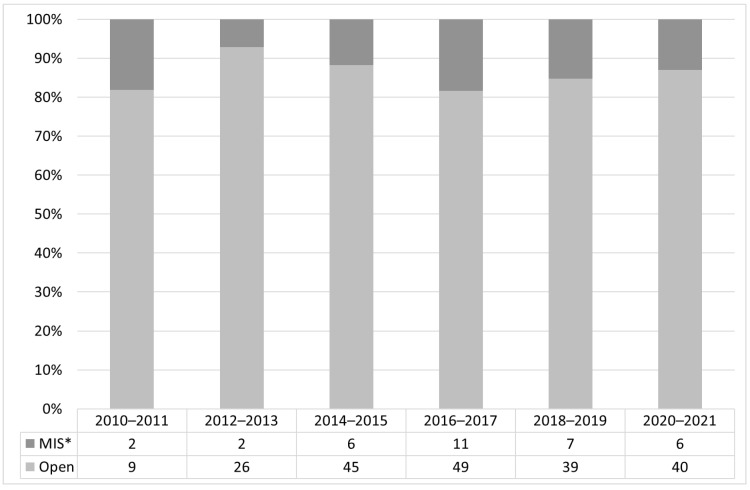
Percentage of minimally invasive vs. open primary anastomosis and tracheoesophageal fistula-repair in the historic control group in patients with esophageal atresia/tracheoesophageal fistula (Gross Type B/C/D, *n* = 242) of the patient registry (courtesy of the German esophageal atresia patient support group KEKS e.V.). *: minimally invasive surgery.

**Figure 2 children-09-00387-f002:**
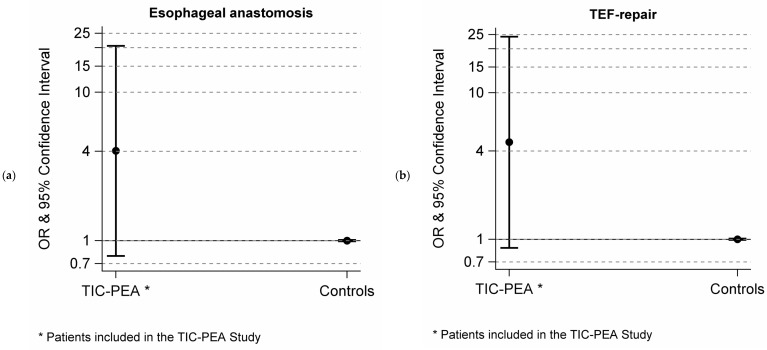
Forest plot of the odds-ratio (OR) on a logarithmic scale of Telemedical Interdisciplinary Care for Patients with Esophageal Atresia (TIC-PEA) patients with with esophageal atresia/tracheoesophageal fistula (Gross Type B/C/D, *n* = 24) with minimally invasive anastomosis (**a**) or trachea-esophageal-fistula (TEF) repair (**b**) compared to controls (*n* = 48) adjusted for birth weight.

**Table 1 children-09-00387-t001:** Comparison of patient characteristics, procedures and complications for TIC-PEA patients and controls with EA/TEF (Gross Type B/C/D) included in the matched-control-analysis.

**Patient Characteristics**	**TIC-PEA (*n* = 24)**	**Control (*n* = 48)**
Mean birth weight (95%-CI) [gr]	2268 (2037–2499)	2330 (2139–2521)
Mean gestational age (95%-CI) [weeks]	35.8 (34.5–37.2)	34.2 (31.5–36.8)
No associated malformation	48%	63%
Congenital heart defect	52%	31%
VACTERL association	20%	0%
**Procedures**	**TIC-PEA (*n* = 24)**	**Control (*n* = 48)**
Mean age primary procedure (95%-CI) [days]	2.3 (1.6–2.9)	2.3 (1.9–2.8)
**Primary anastomosis and TEF closure**	80%	96%
Open	75%	88%
Minimally invasive	25%	10%
**Isolated TEF-repair**	12%	4%
Open	66%	100%
Minimally invasive	33%	0%
**Gastrostomy**	21%	0%
Open	80%	
Minimally invasive	20%	
**Complications**	**TIC-PEA (*n* = 24)**	**Control (*n* = 48)**
No early complications	42%	60%
Anastomotic leakage	13%	6%
Redo-anastomosis	0%	0%
Recurrent fistula	4%	2%
Anastomotic stenosis	38%	10%

**Table 2 children-09-00387-t002:** Comparison of patient characteristics, procedures and complications for TIC-PEA patients and controls with EA (Gross Type A).

**Patient Characteristics**	**TIC-PEA (*n* = 6)**	**Control (*n* = 6)**
Mean birth weight (95%-CI) [gr]	2344 (2065–2635)	1946 (1450–2442)
Mean gestational age (95%-CI) [weeks]	35.6 (33.8–37.5)	38.5 (31.9–39.7)
No associated malformation	50%	33%
Congenital heart defect	50%	66%
VACTERL association	33%	17%
**Procedures**	**TIC-PEA (*n* = 6)**	**Control (*n* = 6)**
Mean age secondary anastomosis (95%-CI) [days]	110.0 (75.6–144.4)	147.3 (87.9–206.6)
**Elongation procedure**	83%	66%
Open	20%	not specified
Minimally invasive	60%	not specified
Endoluminal	20%	75%
**(Partial) gastric transposition**	17%	33%
Open	100%	not specified
Minimally invasive		not specified
**Gastrostomy**	100%	100%
Open	83%	not specified
Minimally invasive	17%	not specified
**Complications**	**TIC-PEA (*n* = 6)**	**Control (*n* = 6)**
No early complications	83%	83%
Anastomotic leakage	17%	17%
Redo-Anastomosis	17%	17%
Anastomotic stenosis	0%	0%

## Data Availability

Restrictions apply to the availability of these data. Data was obtained from KEKS e.V. (Kinder und Erwachsene mit kranker Speiseröhre- the German national patient support group) and are available info@keks.org with the permission of KEKS e.V.

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
