# Peer review of "Telementoring in Minimally Invasive Esophageal Atresia Repair: Results of a Case-Control Study and Lessons Learned from the TIC-PEA Study (Telemedical Interdisciplinary Care for Patients with Esophageal Atresia)"

_children, 2022, doi:10.3390/children9030387_

Round 1

Reviewer 1 Report

Congratulations on this article. I just have some questions that I would like to be answered and included in your manuscript:

In the introduction part, add the meaning of ERNICA (EUROPEAN REFERENCE NETWORK ON RARE INHERITED AND CONGENITAL ANOMALIES)

In the material and methods part:

 -Please describe the expert groups. Were they from the same Country? How many surgeons were in each group?

-What criteria did you follow to decide between open and MIS?

In the results part, please add the mean time of follow up of the patients.

Author Response

Thank you for your comments and work to improve our manuscript. Changes are made and highlighted in the manuscript.

  1. In the introduction part, add the meaning of ERNICA (EUROPEAN REFERENCE NETWORK ON RARE INHERITED AND CONGENITAL ANOMALIES)

Thank you, that must have slipped through. I added the meaning of the abbreviation (line 41)

  1. Please describe the expert groups. Were they from the same Country? How many surgeons were in each group?

Thank you for this comment. I specified in line 66-67 and 72-73: Experts are chairpersons of German academic centers experienced in the treatment of esophageal atresia and/or members of the advisatory board of the KEKS e.V. (Kinder und Erwachsene mit kranker Speiseröhre- the German national patient support group). Individual cases were consulted by the same expert, more or multidisciplinary experts were included, if specific questions arose.

  1. What criteria did you follow to decide between open and MIS?

Most patients had already had surgery before they were enrolled in the study. Therefore, the decision had been made and mentoring consisted in debriefing after surgery. There were no standardized criteria to determine, whether MIS were advisable. Decisions were made individually for each case, depending on the level of the surgeons experience, local infrastructure, type of procedure (esophageal elongation or isolated closure of the fistula, or anastomosis) and patient characteristics (weight, type of esophageal atresia, co-morbidity). We added a clarifying sentence to the manuscript. Line 76-79.

  1. In the results part, please add the mean time of follow up of the patients.

Thank you, I have added the mean follow-up 207.7 days (59-412 days). Line 128-129. By definitiosn, study participation and therefore follow-up ended at the age of one year, leading to a low mean follow-up. Since we were aware of this fact, the study focused on early complications.

Reviewer 2 Report

The authors of this article titled "Telementoring in minimally invasive esophageal atresia repair: 2 results of a case-control study and lessons learned from the 3 TIC-PEA Study (Telemedical Interdisciplinary Care for Patients with Esophageal Atresia).” report their experience with a telemedicine consultation system for the care of patients with esophageal atresia, focusing on his impact in the use of minimally invasive surgery for the treatment of these patients. They compare the data collected over 17 months with the dedicated database of the national patient support group and they conclude that telementoring facilitates minimally invasive repair of esophageal atresia.

Here are my comments:

• Authors should specify the setting of the 16 centers requiring consultation. From the text it appears that these are low volume centers. This analysis is in my opinion essential. In the field of rare conditions, particularly for esophageal atresia, collaboration between highvolume centers is desirable. Telementoring, especially in the management of complex patients, is an objective advantage. However, the possibility for low-volume centers to perform this type of surgery based on telementoring is, in my opinion, ethically questionable.

• Of the 31 patients in the TIC-PEA group, only a small part (23%) was enrolled before surgery.

Only in these cases could telementoring influence the type of surgical approach. Most patients are therefore not fit to meet the purpose of the study

• Authors should specify how 48 controls were extracted from the KEKS group (Page 4 Line

157).

The manuscript is well structured and comprehensible. From this emerges an important role of telemedicine in the management of complex patients suffering from esophageal atresia. However, I have strong ethical concerns that telementoring can replace the experience of a surgeon in a highvolume center. In my opinion this aspect constitutes the major limitation of the manuscript and must be fully discussed in order to be considered for publication.

Author Response

Thank you for your careful review and making us aware, that our massage can be misinterpreted. To avoid misunderstandings we removed the phrase […] telementoring facilitates the use of MIS […] from the conclusion. Further changes are made and highlighted in the manuscript.

  1. Authors should specify the setting of the 16 centers requiring consultation. From the text it appears that these are low volume centers. This analysis is in my opinion essential. In the field of rare conditions, particularly for esophageal atresia, collaboration between highvolume centers is desirable. Telementoring, especially in the management of complex patients, is an objective advantage. However, the possibility for low-volume centers to perform this type of surgery based on telementoring is, in my opinion, ethically questionable.

Thank you for your concern. Unfortunately, the term of high or low-volume center is not well defined for Germany. However, most patients included in the study were treated in academic pediatric surgery units. Indeed, this point is important. We added this information in line 114.

We further like to disagree that telementoring “low-volume centers” to perform this type of surgery is ethically questionable. The participating centers would provide care to these patients independent of the TIC-PEA project. The only difference is that TIC-PEA provides them with feedback on their treatment plan, a means to discuss possible pitfalls, and a way of transfer to a specialized center should this be deemed necessary in the discussion between provider and TIC-PEA specialist.

We have added the following paragraph to the discussion (line 270-280):

“In Germany, the treatment of esophageal atresia is not centralized. Over the period of 2015-2018, 65.8% of patients were treated in small centers (<4 cases/year) and one third (34.2%) in medium centers (4-9 cases/year) – no patient was treated in a high volume center with ten or more cases per year [13]. Nevertheless, the concept of the TIC-PEA study has been controversial among pediatric surgeons in Germany from the beginning. Opponents are concerned, that surgeons with insufficient expertise are encouraged to perform procedures surpassing their own surgical skills instead of transferring patients to more experienced centers. We like to emphasize that telementoring as practiced cannot replace formal surgical training. It aims to support care for patients with rare diseases by providing a national digital network to receive feedback on treatment plans, discuss possible pitfalls or even transfer patients, if deemed necessary.”

  1. Of the 31 patients in the TIC-PEA group, only a small part (23%) was enrolled before surgery. Only in these cases could telementoring influence the type of surgical approach. Most patients are therefore not fit to meet the purpose of the study.

Thank for your comment. Our hypothesis is not, that children in TIC-PEA had more minimally invasive surgery because of TIC-PEA, but that doctors who use minimally invasive procedures profit more from digital consultations, because MIS footage and pictures can be reviewed via telemedicine (As discussed in line 197-202.) For further clarification, we added the following sentence in line 203f.: “Combined with minimally invasive procedures, there is a greater profit from telementoring, compared to open surgery.”

  1. Authors should specify how 48 controls were extracted from the KEKS group (Page 4 Line, 157).

Thank you for your attention, we missed that and specified that controls were randomly selected (Now line 164)

Round 2

Reviewer 2 Report

I have read the edited version of the manuscript. The changes and additions to the text are in my opinion sufficient to clarify my objections. In my opinion, the manuscript is publishable in its current form.